# Glutamine and Cholesterol Plasma Levels and Clinical Outcomes of Patients with Metastatic Castration-Resistant Prostate Cancer Treated with Taxanes

**DOI:** 10.3390/cancers13194960

**Published:** 2021-10-01

**Authors:** Mercedes Marín-Aguilera, María V. Pereira, Natalia Jiménez, Òscar Reig, Anna Cuartero, Iván Victoria, Caterina Aversa, Laura Ferrer-Mileo, Aleix Prat, Begoña Mellado

**Affiliations:** 1Translational Genomics and Targeted Therapeutics in Solid Tumors Group, Institut d’Investigacions Biomèdiques August Pi i Sunyer (IDIBAPS), 08036 Barcelona, Spain; vpereira@oncorosell.com (M.V.P.); najimenez@clinic.cat (N.J.); oreig@clinic.cat (Ò.R.); ivictori@clinic.cat (I.V.); alprat@clinic.cat (A.P.); 2Medical Oncology Department, Hospital Clínic, 08036 Barcelona, Spain; anna_cuartero@hotmail.com (A.C.); aversa@clinic.cat (C.A.); laferrer@clinic.cat (L.F.-M.); 3Rosell Cancer Institute, 08028 Barcelona, Spain; 4Fundació Clínic per a la Recerca Biomèdica, 08036 Barcelona, Spain; 5Department of Medicine, University of Barcelona, 08036 Barcelona, Spain

**Keywords:** biomarkers, prostate cancer, glutamine, liquid biopsy, metabolism, taxanes

## Abstract

**Simple Summary:**

Prostate cancer (PC) is a hormone-dependent disease in which metabolism deregulation has been identified as a relevant event. In particular, glutamine metabolism becomes crucial for biomass and energy production in PC cells. The main aim of this study was to determine whether the plasma glutamine levels correlated with clinical outcomes in patients with metastatic castration resistance prostate cancer (mCRPC) receiving taxanes treatment. We retrospectively assessed the glutamine levels in plasma samples from 75 patients. As glutamine is a precursor of cholesterol production, we also assessed the cholesterol levels in the same cohort, plus 41 extra patients. We found that high glutamine plasma levels were associated with a shorter taxanes response and worse clinical outcome in patients with mCRPC. High cholesterol levels were also indicative of early progression. These results point out circulating glutamine and cholesterol levels as potentially prognostic biomarkers to be further explored in PC.

**Abstract:**

Altered metabolism is a hallmark of cancer. Malignant cells metabolise glutamine to fulfil their metabolic needs. In prostate cancer, androgen receptor signalling promotes glutamine metabolism, which is also involved in cholesterol homeostasis. We aimed to determine whether the plasma glutamine levels correlate with the blood lipid profile, clinical characteristics and outcomes in patients with metastatic castration resistance prostate cancer (mCRPC) undergoing taxanes. We retrospectively assessed the glutamine and glutamate levels in plasma samples by a bioluminescent assay. Pre-treatment glutamine, glutamate, cholesterol and triglycerides levels were correlated with patients’ clinical characteristics, taxanes response and clinical outcomes. Seventy-five patients with mCRPC treated with taxanes were included. The plasma glutamine levels were significantly higher in patients that received abiraterone or enzalutamide prior to taxanes (*p* = 0.003). Besides, patients with low glutamine levels were more likely to present a PSA response to taxanes (*p* = 0.048). Higher glutamine levels were significantly correlated with shorter biochemical/clinical progression-free survival (PSA/RX-PFS) (median 2.5 vs. 4.2 months; *p* = 0.048) and overall survival (OS) (median 12.6 vs. 20.3; *p* = 0.008). High cholesterol levels independently predicted early PSA/RX-PFS (*p* = 0.034). High glutamine and cholesterol in the plasma from patients with mCRPC were associated with adverse clinical outcomes, supporting the relevance of further research on metabolism in prostate cancer progression.

## 1. Introduction

Prostate cancer (PC) is the solid tumour with the second-highest incidence in men [1]. PC is a hormone-dependent disease, in which the activation of the androgen receptor (AR) strongly promotes cancer progression. Androgen deprivation therapy, in combination with novel anti-AR or taxanes, represents a fundamental strategy for the treatment of hormone-sensitive [2,3,4,5] and metastatic castration-resistant PC (mCRPC) [5,6,7]. Despite significant therapeutic achievements in the last years, mCRPC remains an aggressive and incurable disease.

Metabolism deregulation has been identified as a relevant event in cancer progression [8]. Tumour cells present abundant energetic and biosynthesis requirements, which lead to the emergence of metabolic alterations to sustain their growth [9]. In particular, the glutamine metabolism pathway is crucial for protein/lipid synthesis, energy production and nitrogen/carbon sourcing in cancer cells. It is catabolised to glutamate and α-ketoglutarate, which is incorporated into the tricarboxylic acid cycle to produce energy. Besides, glutamine is the precursor for a number of biosynthetic pathways required for growth and cell division [10].

In PC, multiple metabolic abnormalities, including lipid and amino acids pathways, have been reported and are associated with PC progression and/or the development of mCRPC [11,12]. Furthermore, the metabolism of glutamine may be promoted by AR signalling and is upregulated in PC [12]. In addition, it is regulated by MYC oncogene [13] and has been associated with neuroendocrine differentiation [14]. Importantly, glutamine is a precursor of fatty acids and cholesterol, which have also been described as a relevant deregulated metabolic pathway in PC [15]. The involvement of lipids in PC progression is not yet fully understood, but it is known that androgens promote lipid synthesis through a gene expression program that remains functional in mCRPC [16].

As glutamine may be detected in plasma, we analysed whether blood plasma glutamine correlated with the lipid profile and whether it may influence clinical outcomes of patients with mCRPC treated with taxanes.

## 2. Materials and Methods

### 2.1. Study Design

We prospectively collected blood samples from patients with mCRPC receiving treatment with docetaxel or cabazitaxel as part of a study carried out in our institution that investigated blood biomarkers in patients receiving different antitumour therapies. Additionally, we collected blood test laboratory data from a set of patients with mCRPC treated with taxanes as a validation cohort. The hospital’s Institutional Ethics Committee approved the study, and all participants provided written informed consent. This study followed the REMARK recommendations for tumour marker prognostic studies [17].

The patients were treated with docetaxel (75 mg/m^2^) or cabazitaxel (25 mg/m^2^) intravenously every 3 weeks. The treatment–response criteria and progressive-disease definitions followed the Prostate Cancer Working Group 2 criteria [18]. For each patient included, we collected clinical, pathological and laboratory data, including cholesterol and triglycerides levels, prior to the start of taxanes.

### 2.2. Samples Collection and Glutamine Determination

Peripheral blood samples were collected in 10-mL EDTA-containing tubes and kept at 4 °C for up to 2 h until processed. Plasma was obtained by the centrifugation of samples at 560× *g* for 10 min and was stored in 1-mL aliquots at −80 °C until used. Plasma samples were used at a 1:50 dilution in phosphate-buffered saline solution for glutamine and glutamate level determination, which were evaluated by using the Glutamine/Glutamate-GloTM kit (J8022, Promega, Madison, WI, USA) following the manufacturer’s instructions. Assays were performed blinded to the study endpoint.

### 2.3. Statistical Analysis

Fisher’s exact and Wilcoxon Mann–Whitney tests were used to compare the proportions in the qualitative and continuous variables, respectively. Optimal cut-offs for the glutamine and glutamate levels were assessed using maximally selected log-rank statistics (Maxstat package) [19] to segregate patients in low- and high-level groups according to the primary endpoints: biochemical progression-free survival (PSA-PFS), first progression (PSA/RX-PFS) and overall survival (OS). PSA-PFS, PSA/RX-PFS and OS were calculated from the date of taxanes initiation to PSA progression, biochemical/clinical/radiological progression and death or the last follow-up visit, respectively. The first progression variable integrated both biochemical and clinical/radiological PFS, taking into account the one that occurred earlier. The cut-offs values for the analytical parameters included in the analysis were the ones established as a reference in our hospital (i.e., 247 mg/dL and 150 mg/dL for cholesterol and triglycerides, respectively).

The survival analysis was evaluated by a log-rank test. Correlations between continuous variables were measured by calculating the Pearson’s coefficient. A univariate analysis was performed by Cox regression; a *p*-value (*p*) < 0.1 was required for inclusion in the multivariate analysis. The complete cases and regression analysis allowed dealing with missing data. All tests were 2-sided, and *p* < 0.05 was considered significant. Statistical analysis was performed with R software (R project, Vienna, Austria) (v.3.6.3)) [20].

## 3. Results

### 3.1. Patient Characteristics

From September 2011 to November 2018, 75 patients were included in the study, and 85 plasma samples were collected before the initiation of taxanes. The median age was 70.4 (range 55.8–83.5) years, and the median follow-up was 13.8 (range 1–53.9) months. Among them, 57 (67.1%) and 28 (32.9%) samples were collected prior to receiving docetaxel and cabazitaxel treatments, respectively. Ten (13.3%) patients received both treatments, and two samples per patient were included in the study (Figure 1). During the follow-up time, 79 PSA progression events, 85 when combining the PSA and radiological criteria and 66 deaths were observed. From 35 patients, we also collected post-treatment samples, 29 (82.9%) after docetaxel and 11 (31.4%) after cabazitaxel, summing a total of 76 samples in the cohort. Five patients received both docetaxel and cabazitaxel, and for four of them, the same post-docetaxel sample served as the pre-cabazitaxel one, allowing the study of 40 paired cases (Figure 1). We also collected blood test records from a cohort of 41 patients (44 blood tests) who received taxanes in our institution, 36 treated with docetaxel and eight with cabazitaxel, as a validation cohort for the lipid parameters. The clinical characteristics of all the included patients are shown in Table 1.

### 3.2. Glutamine Plasma Levels and Clinical Outcomes

Glutamine was detected in all the analysed samples (mean 414 μM, range 194.2–772.1 μM). There was no correlation between the glutamine levels and Gleason score, pre-taxanes serum levels of lactate dehydrogenase (LDH), haemoglobin (Hb), PSA and alkaline phosphatase or the metastasis location (presence/absence of bone or visceral) pre-treatment. Patients who received treatment with abiraterone or enzalutamide prior to taxanes (*N* = 42 and 56%) had significantly higher plasma glutamine levels (*p* = 0.003) and its derivate glutamate (*p* = 0.026) compared to those who did not receive those therapies prior to the taxanes (*N* = 33 and 44%) (Figure 2A).

By establishing a cut-off for the glutamine levels according to PSA-PFS, 44 (51.8%) and 41 (48.2%) samples were categorised as having low and high glutamine contents, respectively. The median follow-up time for patients with low and high glutamine contents was 17.8 (range 3.3–53.9) and 12.6 (range 1–35.9) months, respectively. More patients responded to the taxanes in the group of low glutamine (64.8%) compared to the group of high glutamine levels (35.2%) (*p* = 0.048) (Figure 2B,C). Patients with high glutamine levels had a shorter time to progression (median 2.5 vs. 4.2 months, HR 1.6, 95%CI 1–2.5, *p* = 0.048) and shorter OS (median 12.6 vs. 20.3 months, HR 2, 95%CI 1.2–3.2, *p* = 0.008) than those with low glutamine levels. No significant differences were found when considering only the biochemical progression (PSA-PFS) (Figure 2D) or in the multivariate analysis (Table 2). Of note, the patients that received abiraterone or enzalutamide prior to taxanes had a shorter time to progression (median 2.03 vs. 4.2 months, HR 1.6, 95%CI 1–2.5, *p* = 0.048) and shorter OS (median 10.3 vs. 12 months, HR 1.7, 95%CI 1–2.7, *p* = 0.032) than those who did not receive such treatments before the taxanes (Appendix A). Receiving abiraterone or enzalutamide prior to the taxanes was independently associated with a worse OS (median 12.7 vs. 22 months, HR 1.9, 95%CI 1.02–3.4, *p* = 0.043) (Table 2). No significant interaction between the glutamine levels and receiving abiraterone or enzalutamide before the taxanes to the PSA-PFS, PSA/RX-PFS or OS was found (*p* = ns).

As a direct metabolite of glutamine production/consumption, we also determined the glutamate levels in the plasma samples and their correlation with the clinical parameters before taxane exposure in the same cohort of patients. The glutamine and glutamate plasma levels were positively correlated (*p* = 0.033) (Figure 3), but neither the pre-taxanes glutamate levels nor the glutamine/glutamate ratio correlated with the clinical outcomes (Appendix A).

### 3.3. Glutamine Analysis in the Sequential Samples

To seek whether glutamine and glutamate levels were modulated by the taxanes treatment, we determined their levels in 40 paired pre- and post-taxane samples. This analysis did not result in significant glutamine levels changes after taxane exposure, while glutamate levels were significantly higher in the post-taxane samples (*p* < 0.001); hence, the ratio glutamine/glutamate was significantly reduced (*p* = 0.004) (Figure 4). None of the changes in these three parameters after the taxane exposure was associated with clinical outcomes (Appendix A).

### 3.4. Glutamine and Lipids

Since glutamine is a precursor of fatty acids, we explored whether glutamine and/or glutamate plasma levels correlated with the blood circulating cholesterol and triglycerides levels. No significant correlation among these variables was found (Appendix A). However, high cholesterol levels before the start of the taxanes were associated with a shorter PSA-PFS (median 3 vs. 4.6 months, HR 2.4; 95% CI 1.1–5.2; *p* = 0.026) but not with OS (Figure 5A). No significant differences were found in the multivariate analysis (Table 3). These results were confirmed in a validation cohort of 41 patients (44 blood tests) also treated with taxanes in our institution. The patients’ characteristics of the validation cohort are shown in Table 1. Patients with high cholesterol levels had shorter PSA-PFS (median 2.7 vs. 5.2 months, HR 3.3; 95% CI 1.2–9; *p* = 0.019) and PSA/RX-PFS (median 2.7 vs. 4.8 months, HR 2.8; 95% CI 1.7–5; *p* = 0.04) than the patients with low cholesterol in the blood (Figure 5B). The analysis of both cohorts together (referred to as the global cohort) also showed that high cholesterol was an independent predictor of shorter PSA-PFS (median 2.7 vs. 4.7 months, HR 2.5; 95% CI 1.2–4.9; *p* = 0.012) and PSA/RX-PFS (median 2.6 vs. 4.1 months, HR 2.1; 95% CI 1.1–4.2; *p* = 0.034) (Table 4 and Appendix A). Of note, no significant interaction between the glutamine or glutamate expression levels and circulating cholesterol related to PSA-PFS was found (*p* = 0.586). No significant association of triglycerides with the clinical outcomes was found either (Appendix A).

We also explored whether the treatment with statins, which reduce the cholesterol accumulation in the blood, affected the clinical outcomes and plasma glutamine levels. No association between the receiving statins and clinical outcomes was observed, although the patients receiving statin had higher levels of glutamine (*p* = 0.009) (Figure 6).

## 4. Discussion

Cancer—and, in particular, PC cells—usually resort to the use of glutamine as an extra source of energy for proliferation, growth and metastatic expansion. To our knowledge, circulating plasma glutamine as an indicator of altered metabolism has not yet been explored in the literature as a biomarker of tumour evolution and/or treatment response, and it has been addressed in the present study.

The main finding of this work is that high cholesterol levels are independently correlated with shorter PSA-PFS and PSA/RX-PFS to taxanes in patients with mCRPC. Furthermore, high glutamine levels in plasma are associated with earlier progression and shorter OS. These results suggest a more aggressive behaviour of tumours with active cholesterol and glutamine metabolisms.

The association of glutamine and taxanes responses was preclinically supported by the work of Ippolito et al., who showed that PC3 cell lines resistant to docetaxel exhibit higher glutamine-dependent growth compared to parental cells and that the inhibition of glutaminase affects their ability to invade [21]. Moreover, we observed a significant increase of the glutamate plasma levels after receiving taxanes treatment, which could be interpreted as an increase in glutamine metabolization by the organism. However, the fact that glutamine is not significantly modified could indicate a potential activation of de novo glutamine synthesis in order to achieve the extra energy and biomass required by tumour cells during chemotherapy treatment. Further experimental data would contribute to demonstrating such a hypothesis.

It has been postulated that metabolic reprogramming becomes a key factor as tumours progress [22], enabling tumours to adapt to serial therapies. In this context of the advanced disease, we observed higher levels of glutamine in the plasma from patients that received abiraterone or enzalutamide prior to taxanes. This is consistent with the observation that androgen deprivation induces glutamine accumulation produced by PC cells and with metabolic reprogramming as the tumour deals with the treatment [23]. Moreover, PC patients with therapeutic resistance to androgen deprivation have a higher number of metabolic alterations compared to those with a responsive disease [24]. Of note, AR directly regulates the glutamine pathway by upregulating the glutamine utilisation and the expression of the glutamine transporters [12,23]. Indeed, antagonizing the uptake of glutamine through metabotropic glutamate receptor 1 antagonists has been shown to restore sensitivity to AR inhibition in preclinical models [14]. Glutamine metabolism is also regulated by the MYC oncogene [13], which is frequently upregulated in PC and also promotes the transcription of AR [25] and is associated with neuroendocrine dedifferentiation and mCRPC progression [26]. Glutamine deprivation in MYC-overexpressing cancer cells results in MYC-dependent apoptosis [27]. Besides, it has been reported in preclinical models that stromal glutamine serves as a mediator of PC neuroendocrine differentiation [14], which has been related to both hormone therapy and taxanes resistance. Moreover, glutamine is an important signalling molecule that has been implicated in the activation of the mammalian target of rapamycin (mTOR), stimulation of protein synthesis, cell growth and differentiation and the inhibition of protein degradation and apoptosis [28]. Altogether, it seems clear that AR modulates the glutamine metabolism as an interesting pathway to be targeted in PC (Figure 7).

The dependency of tumour cells toward glutamine metabolism points out this pathway as a potential target for treatment, as it has been tested in several studies. A novel small molecule that inhibits glutaminase isoforms not commonly expressed in normal cells [29] is under study in two phase II trials (NCT03163667 and NCT03428217) for advanced renal cell carcinoma. It is also being tested in other cancer types [30]. Another recent study revealed that the inhibition of guanosine monophosphate synthetase, which uses glutamine as a source of nitrogen to synthesise guanine, decreased the tumour growth in PC-3 xenograft models [31].

Glutamine metabolism has been also demonstrated to be a key pathway to sustain immune T-cell population proliferation [32]. Recently, a preclinical study showed that a glutamine antagonist, 6-diazo-5-oxo-L-norleucine, was able to modulate tumour and immune cell metabolism, favouring the activation of the immune system against tumours and achieving an inhibition of proliferation and cell viability [33]. Although PC was not explored in that work, such results deserve to be tested in PC, where the immunogenicity is not abundant, as a strategy to reinforce the immune cells activity within the antitumoural scene.

We also explored the association of glutamine plasma levels with other lipid metabolic parameters in the blood. Hypercholesterolemia has been described as a risk factor for developing aggressive PC. Indeed, cholesterol may act as a direct precursor of androgen synthesis in PC, becoming an essential promoter of cancer cell proliferation [34]. Reciprocally, androgen signalling upregulates the expression of enzymes involved in the endogenous synthesis of lipids, such as fatty acid synthase [35]. Although glutamine is a precursor of fatty acids and cholesterol production, we did not observe a correlation between their levels in the plasma, and no significant interaction was found between them.

Our results also suggested a detrimental effect of high cholesterol levels in clinical outcomes of patients treated with taxanes. It has been suggested that statins may have a positive impact on PC progression by lowering the blood cholesterol, a substrate for adrenal and intratumoural androgen biosynthesis. Di Lorenzo et al., in a retrospective study, showed that receiving statins was a significant prognostic factor for longer OS and produced significantly higher PSA decline rates in mCRPC patients treated with abiraterone [36]. A post hoc analysis of two prospective randomised clinical trials associated statin use with a superior OS in patients with mCRPC treated with prednisone or abiraterone/prednisone [37]. In our series, receiving statins was not associated with better outcomes, but it is of note that patients receiving statins had higher glutamine levels. It has been reported that statins may modulate the glutamine metabolism in neurological disorders. Essentially, they are able to increase the glutamine synthetase activity by promoting extracellular glutamate uptake, which reduces oxidative stress in brain cells [38,39]. Interestingly, it has been described that the administration of metformin to reduce the glucose levels increases the glutamine dependency of tumour cells, suggesting a potential synergism of metformin in combination with glutamine pathway inhibitors [40]. The modulation of the glutamine metabolism through statins specifically in PC has not been investigated to date, and further studies are required to confirm whether the use of statins concomitantly with glutamine blockers agents could benefit PC patients.

The main limitation of this work relied on the lack of an independent validation of the plasma level-related results. However, due to the novelty character of the findings, and the potential interest that they could arise in other groups, we presented these results while we increased the number of plasma samples of patients undergoing different treatments for further validation.

## 5. Conclusions

In conclusion, this study provided evidence about the potential role of the cholesterol and glutamine metabolic pathways in treatment resistance, progression and as new therapeutic targets to be further explored in mCRPC. Further research to validate these results needs to be conducted.

## Figures and Tables

**Figure 1 cancers-13-04960-f001:**
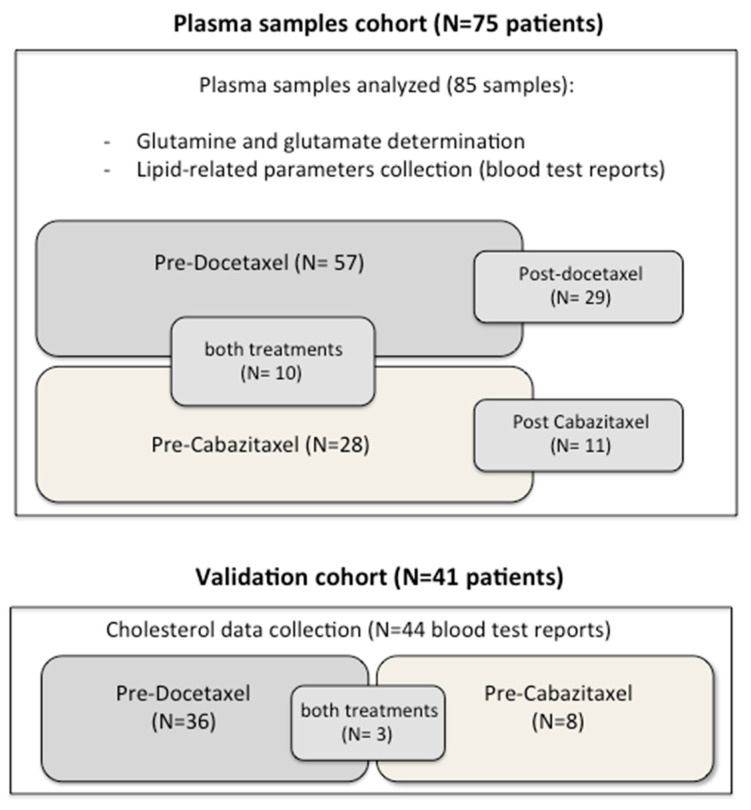
Scheme of the patients included in this study. *N*: number of patients.

**Figure 2 cancers-13-04960-f002:**
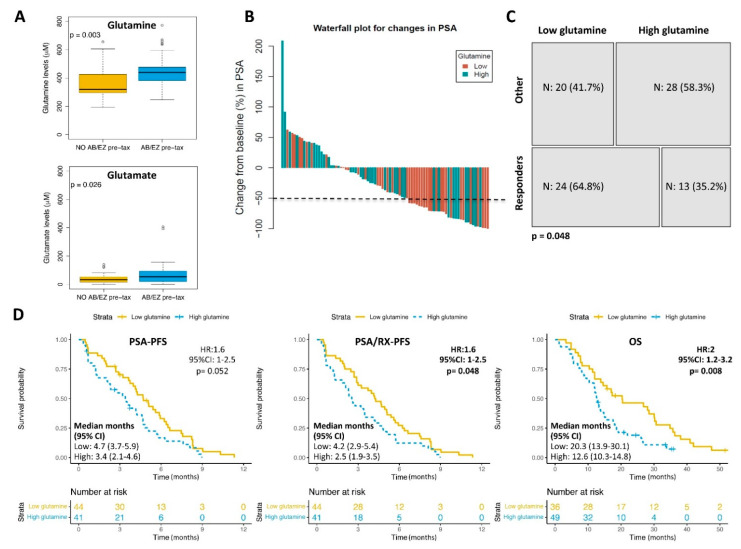
Plasma levels of glutamine and glutamate and the clinical outcome. (**A**) Boxplot of the glutamine (up) and glutamate (down) levels in patients according to the administration of hormone therapy with abiraterone (AB) or enzalutamide (EZ) before taxanes. *t*-test (*p*-value). (**B**) Waterfall plot showing the % of change in the prostatic-specific antigen (PSA) levels after taxanes treatment according to the plasma glutamine levels (high or low). The dotted line represents the limit to consider the biochemical response. (**C**) Contingency table showing the number (*N*) and % of patients responding to the taxanes vs. non-responders or the stable disease (other), according to the glutamine levels. Fisher’s Exact Test (*p*-value). (**D**) Kaplan–Meier curve representing PSA-PFS, PSA/RX-PFS and OS according to the glutamine plasma levels. CI: confidence interval; HR: hazard ratio; Wald test (*p*-value); Time in months.

**Figure 3 cancers-13-04960-f003:**
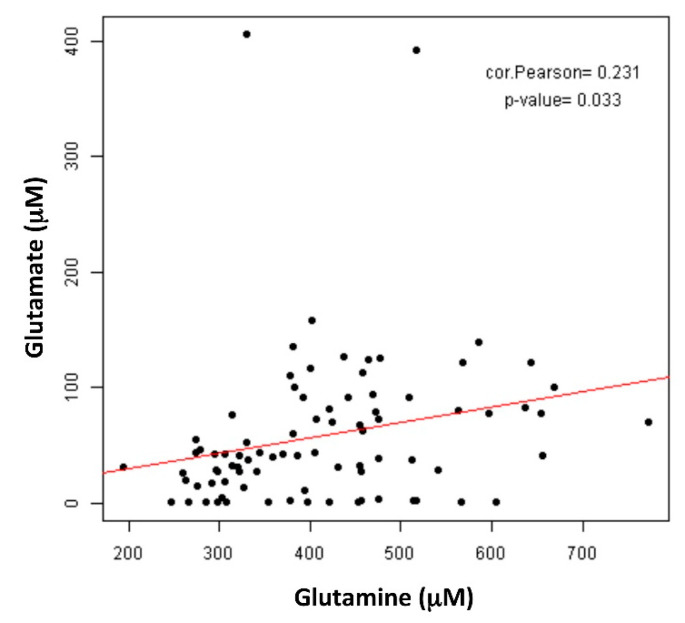
Correlation between glutamine and glutamate plasma levels. Pearson’s coefficient (cor) and *p*-value are shown.

**Figure 4 cancers-13-04960-f004:**
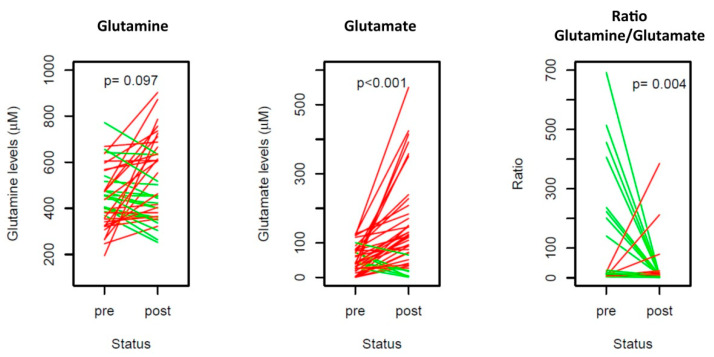
Changes in glutamine, glutamate and their ratio levels from the start of the taxanes treatment (pre) to after the treatment (post). Levels going up and down are represented in red and green, respectively. The Wilcoxon signed-rank test *p*-value (*p*) is shown.

**Figure 5 cancers-13-04960-f005:**
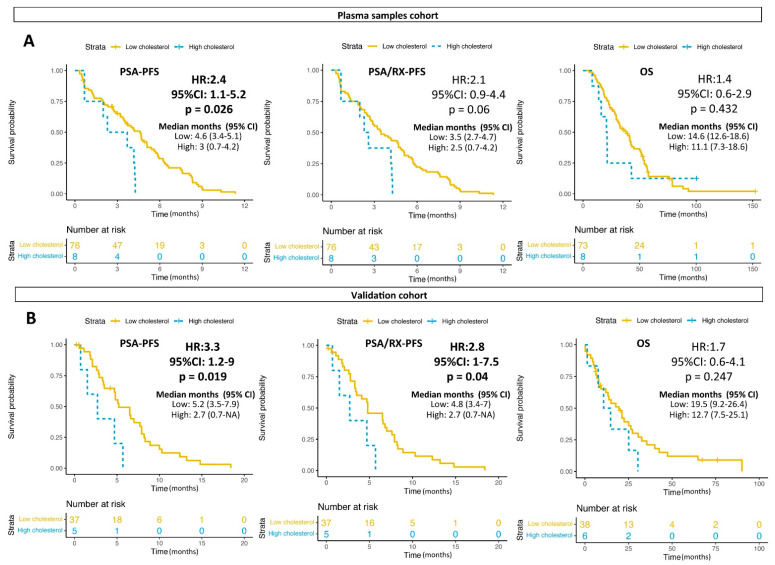
Cholesterol levels and clinical outcomes. (**A**) Kaplan–Meier curves representing PSA-PFS, PSA/RX-PFS and OS according to the cholesterol levels in the plasma samples cohort. (**B**) Kaplan–Meier curves representing PSA-PFS, PSA/RX-PFS and OS according to cholesterol levels in a validation cohort. Cut-off for the cholesterol levels was the one used at the routine blood test analysis (247 mg/dL). CI: confidence interval; HR: hazard ratio; Wald test (*p*); Time is in months.

**Figure 6 cancers-13-04960-f006:**
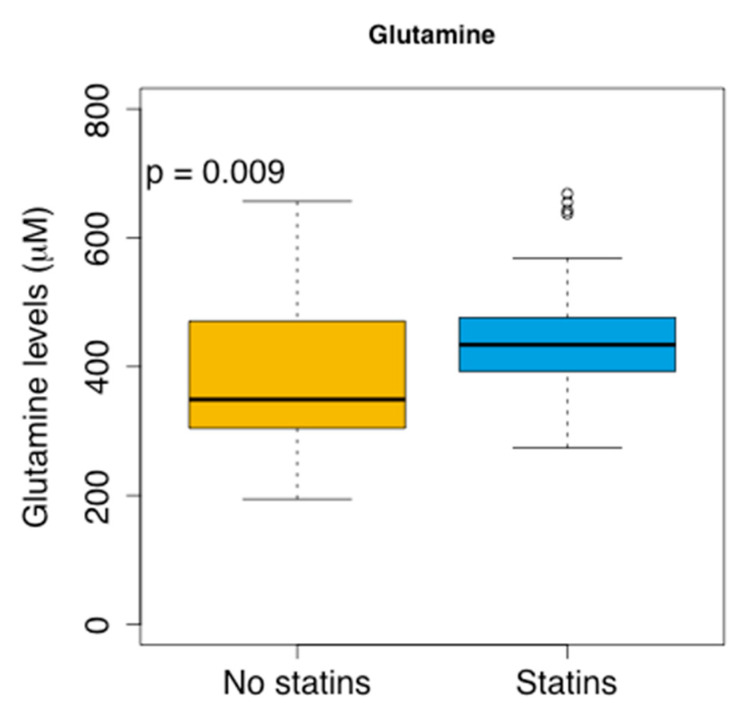
Glutamine levels according to the administration of the statins. *t*-test (*p*-value).

**Figure 7 cancers-13-04960-f007:**
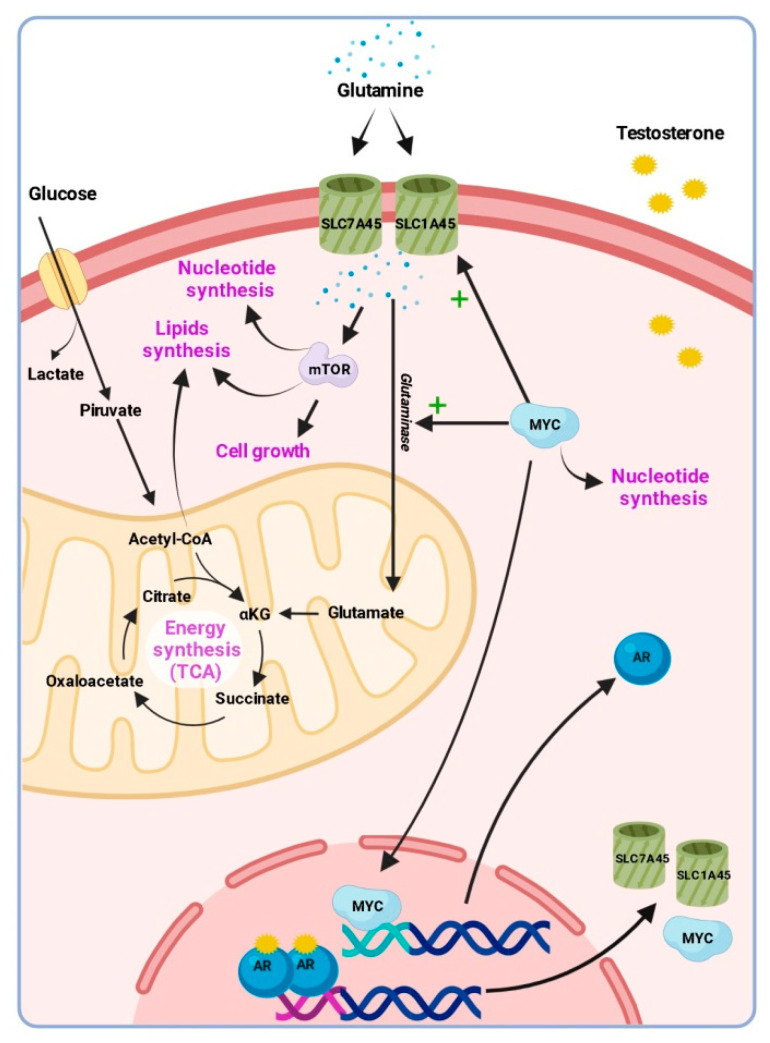
A scheme representing the relationship between glutamine metabolism and androgen receptor (AR) signalling. AR upregulates the expression of glutamine transporters and the MYC oncogene, which also promotes the transcription of AR. MYC also activates glutamine transport and glutaminase activity, which convert glutamine to glutamate and enters the tricarboxylic acid cycle (TCA) to synthesise energy. Glutamine metabolism is also involved in lipids synthesis. Glutamine also participates in the activation of the mammalian target of rapamycin (mTOR), which stimulates lipid and nucleotide synthesis, cell growth and differentiation.

**Table 1 cancers-13-04960-t001:** Patients’ characteristics.

Characteristics	Plasma Samples Cohort	Cholesterol Validation Cohort
Number of patients (samples)	75 (85)	41 (44)
Age at treatment (years)		
Median (range)	70.4 (55.8–83.5)	71.6 (37.3–85.9)
Time since diagnosis to treat (years)		
Median (range)	4.5 (0.34–21.5)	4.7 (0.33–14.01)
Stage at diagnosis, *N* (%)		
≤3	33 (44)	15 (36.6)
≥4	36 (48)	25 (60.9)
NA	6 (8)	1 (2.4)
Gleason sum at diagnosis, *N* (%)		
≤7	28 (37.3)	14 (34.1)
≥8	45 (60)	23 (9.8)
NA	2 (2.7)	4 (9.8)
Presence of bone metastases, *N* (%)		
Yes	69 (92)	38 (92.7)
No	6 (8)	6 (14.6)
Presence of visceral metastases, *N* (%)		
Yes	16 (21.3)	15 (36.6)
No	59 (78.7)	26 (63.4)

ECOG performance status score, *N* (%)		
0	12 (16)	9 (21.9)
1 or 2	61 (81.3)	31 (75.6)
NA	2 (2.7)	1 (2.4)
Baseline Prostate-specific antigen (ng/mL)		
Median (range)	42.17 (0.04–2398.7)	42.41 (0.04–1284)
Baseline hemoglobin concentration (g/L)		
Median (range)	124.1 (87–154)	121.5 (15.3–151)
Baseline alkaline phosphatase (U/L)		
Median (range)	180 (54–1953)	191.5 (75–2311)
Baseline lactate dehydrogenase (U/L)		
Median (range)	378 (151–2381)	381.5 (163–1979)
Use of abiraterone/enzalutamide, *N* (%)		
Pre-chemotherapy	42 (56)	25 (60.9)
Never or Post-chemotherapy	33 (44)	16 (39)

*N*: number of cases; ECOG: Eastern Cooperative Oncology Group.

**Table 2 cancers-13-04960-t002:** Univariate and multivariate Cox model for PSA-PFS, PSA/RX-PFS and OS in patients from the plasma samples cohort adjusted for clinically significant variables (*p* < 0.1) in the univariate analysis.

	Univariate				Multivariate			
Variable	HR	95% CI		*p*-Value	HR	95% CI		*p*-Value
**PSA-PFS**								
Stage at diagnosis *	1.04	0.65	1.65	0.880	-	-	-	-
Gleason at diagnosis *	1.43	0.90	2.30	0.134	-	-	-	-
ECOG *	1.60	0.81	3.16	0.179	-	-	-	-
LDH **	1.00	1.00	1.00	0.006	1.00	1.00	1.00	0.012
Hb **	0.98	0.96	1.00	0.016	0.98	0.97	1.00	0.026
PSA **	1.00	1.00	1.00	0.010	1.00	1.00	1.00	0.036
AP **	1.00	1.00	1.00	0.862	-	-	-	-
Visceral metastases *	0.90	0.52	1.58	0.724	-	-	-	-
Bone metastases *	0.56	0.24	1.31	0.182	-	-	-	-
AB/EZ prior to taxanes *	1.59	1.00	2.52	0.048	1.67	0.98	2.84	0.058
Glutamine levels *	1.57	1.00	2.46	0.052	1.28	0.77	2.15	0.344
**PSA/RX-PFS**								
Stage at diagnosis *	1.09	0.69	1.70	0.709	-	-	-	-
Gleason at diagnosis *	1.57	0.99	2.47	0.053	1.78	1.01	2.91	0.022
ECOG *	1.61	0.84	3.07	0.152	-	-	-	-
LDH **	1.00	1.00	1.00	0.009	1.00	1.00	1.00	0.108
Hb **	0.98	0.96	0.99	0.007	0.98	0.96	0.99	0.006
PSA **	1.00	1.00	1.00	0.002	1.00	1.00	1.00	0.009
AP **	1.00	1.00	1.00	0.608	-	-	-	-
Visceral metastases *	0.96	0.57	1.63	0.894	-	-	-	-
Bone metastases *	0.56	0.25	1.22	0.145	-	-	-	-
AB/EZ prior to taxanes *	1.48	0.95	2.29	0.081	1.42	0.83	2.43	0.198
Glutamine levels *	1.55	1.00	2.40	0.049	1.47	0.88	2.48	0.144
**OS**								
Stage at diagnosis *	0.98	0.61	1.58	0.932	-	-	-	-
Gleason at diagnosis *	1.52	0.94	2.47	0.087	1.69	1.00	2.85	0.049
ECOG *	1.60	0.82	3.13	0.170	-	-	-	-
LDH **	1.00	1.00	1.00	0.000	1.00	1.00	1.00	0.011
Hb **	0.97	0.95	0.99	0.001	0.97	0.95	0.99	0.001
PSA **	1.00	1.00	1.00	0.091	1.00	1.00	1.00	0.241
AP **	1.00	1.00	1.00	0.775	-	-	-	-
Visceral metastases *	1.01	0.60	1.72	0.964	-	-	-	-
Bone metastases *	0.84	0.38	1.83	0.657	-	-	-	-
AB/EZ prior to taxanes *	1.69	1.05	2.72	0.032	1.86	1.02	3.41	0.043
Glutamine levels *	1.95	1.19	3.21	0.009	1.49	0.83	2.69	0.183

* Variables considered dichotomic. ** Variables considered continuous. ECOG: Eastern Cooperative Oncology Group; PSA: prostate-specific antigen; Hb: haemoglobin concentration; LDH: lactate dehydrogenase; AP: alkaline phosphatase; AB: Abiraterone; EZ: Enzalutamide; HR: hazard ratio; CI: confidence interval.

**Table 3 cancers-13-04960-t003:** Univariate and multivariate Cox models for PSA-PFS, PSA/RX-PFS and OS in patients from the plasma sample cohort adjusted for clinically significant variables (*p* < 0.1) in the univariate analysis.

	Univariate				Multivariate			
Variable	HR	95% CI		*p*-Value	HR	95% CI		*p*-Value
**PSA-PFS**								
Stage at diagnosis *	1.04	0.65	1.65	0.880	-	-	-	-
Gleason at diagnosis *	1.43	0.90	2.30	0.134	-	-	-	-
ECOG *	1.60	0.81	3.16	0.179	-	-	-	-
LDH **	1.00	1.00	1.00	0.006	1.00	1.00	1.00	0.017
Hb **	0.98	0.96	1.00	0.016	0.98	0.97	1.00	0.052
PSA **	1.00	1.00	1.00	0.010	1.00	1.00	1.00	0.024
AP **	1.00	1.00	1.00	0.862	-	-	-	-
Visceral metastases *	0.90	0.52	1.58	0.724	-	-	-	-
Bone metastases *	0.56	0.24	1.31	0.182	-	-	-	-
AB/EZ prior to taxanes *	1.59	1.00	2.52	0.048	1.91	1.12	3.19	0.009
Cholesterol levels *	2.39	1.11	5.15	0.026	2.10	0.92	4.78	0.078
**PSA/RX-PFS**								
Stage at diagnosis *	1.09	0.69	1.70	0.709	-	-	-	-
Gleason at diagnosis *	1.57	0.99	2.47	0.053	1.61	0.97	2.70	0.068
ECOG *	1.61	0.84	3.07	0.152	-	-	-	-
LDH **	1.00	1.00	1.00	0.009	1.00	1.00	1.00	0.121
Hb **	0.98	0.96	0.99	0.007	0.98	0.96	1.00	0.013
PSA **	1.00	1.00	1.00	0.002	1.00	1.00	1.00	0.005
AP **	1.00	1.00	1.00	0.608	-	-	-	-
Visceral metastases *	0.96	0.57	1.63	0.894	-	-	-	-
Bone metastases *	0.56	0.25	1.22	0.145	-	-	-	-
AB/EZ prior to taxanes *	1.48	0.95	2.29	0.081	1.75	1.09	2.82	0.021
Cholesterol levels *	2.06	0.96	4.39	0.062	1.42	0.59	3.38	0.426
**OS**								
Stage at diagnosis *	0.98	0.61	1.58	0.932	-	-	-	-
Gleason at diagnosis *	1.52	0.94	2.47	0.087	1.65	0.98	2.78	0.058
ECOG *	1.60	0.82	3.13	0.170	-	-	-	-
LDH **	1.00	1.00	1.00	0.000	1.00	1.00	1.00	0.011
Hb **	0.97	0.95	0.99	0.001	0.96	0.94	0.98	0.000
PSA **	1.00	1.00	1.00	0.091	1.00	1.00	1.00	0.229
AP **	1.00	1.00	1.00	0.775	-	-	-	-
Visceral metastases *	1.01	0.60	1.72	0.964	-	-	-	-
Bone metastases *	0.84	0.38	1.83	0.657	-	-	-	-
AB/EZ prior to taxanes *	1.69	1.05	2.72	0.032	2.28	1.34	3.87	0.002
Cholesterol levels *	1.37	0.63	3.00	0.432	-	-	-	-

* Variables considered dichotomic. ** Variables considered continuous. ECOG: Eastern Cooperative Oncology Group; PSA: prostate-specific antigen; Hb: haemoglobin concentration; LDH: lactate dehydrogenase; AP: alkaline phosphatase; HR: hazard ratio; CI: confidence interval.

**Table 4 cancers-13-04960-t004:** Univariate and multivariate Cox models for PSA-PFS, PSA/RX-PFS and OS in patients from the plasma samples cohort, together with the validation cohort (global cohort), adjusted for clinically significant variables (*p* < 0.1) in the univariate analysis.

	Univariate				Multivariate			
Variable	HR	95% CI		*p*-Value	HR	95% CI		*p*-Value
**PSA-PFS**								
Stage at diagnosis *	0.91	0.62	1.33	0.634	-	-	-	-
Gleason at diagnosis *	1.41	0.95	2.09	0.085	1.11	0.72	1.70	0.643
ECOG *	1.76	1.04	2.98	0.035	1.59	0.92	2.74	0.096
LDH **	1.00	1.00	1.00	0.001	1.00	1.00	1.00	0.004
Hb **	1.00	0.99	1.01	0.943	-	-	-	-
PSA **	1.00	1.00	1.00	0.003	1.00	1.00	1.00	0.040
AP **	1.00	1.00	1.00	0.212	-	-	-	-
Visceral metastases *	0.77	0.51	1.16	0.209	-	-	-	-
Bone metastases *	0.89	0.47	1.67	0.715	-	-	-	-
AB/EZ prior to taxanes *	1.14	0.78	1.66	0.507	-	-	-	-
Cholesterol levels *	2.54	1.39	4.65	0.002	2.45	1.22	4.92	0.012
**PSA/RX-PFS**								
Stage at diagnosis *	0.93	0.64	1.34	0.688	-	-	-	-
Gleason at diagnosis *	1.58	1.08	2.31	0.020	1.31	0.86	1.99	0.210
ECOG *	1.84	1.10	3.06	0.019	1.63	0.96	2.77	0.069
LDH **	1.00	1.00	1.00	0.001	1.00	1.00	1.00	0.004
Hb **	1.00	0.99	1.01	0.884	-	-	-	-
PSA **	1.00	1.00	1.00	0.001	1.00	1.00	1.00	0.010
AP **	1.00	1.00	1.00	0.424	-	-	-	-
Visceral metastases *	0.83	0.56	1.24	0.363	-	-	-	-
Bone metastases *	0.89	0.49	1.63	0.703	-	-	-	-
AB/EZ prior to taxanes *	1.06	0.734	1.52	0.772	-	-	-	-
Cholesterol levels *	2.16	1.19	3.93	0.011	2.12	1.06	4.22	0.034
**OS**								
Stage at diagnosis *	1.03	0.71	1.51	0.874	-	-	-	-
Gleason at diagnosis *	1.48	1.00	2.20	0.053	1.41	0.92	2.15	0.114
ECOG *	1.77	1.04	3.01	0.037	1.76	1.00	3.13	0.052
LDH **	1.00	1.00	1.00	0.000	1.00	1.00	1.00	0.000
Hb **	1.00	1.00	1.01	0.444	-	-	-	-
PSA **	1.00	1.00	1.00	0.118	-	-	-	-
AP **	1.00	1.00	1.00	0.248	-	-	-	-
Visceral metastases *	0.88	0.58	1.33	0.541	-	-	-	-
Bone metastases *	1.01	0.56	1.84	0.970	-	-	-	-
AB/EZ prior to taxanes *	1.21	0.83	1.77	0.315	-	-	-	-
Cholesterol levels *	1.49	0.83	2.67	0.181	-	-	-	-

* Variables considered dichotomic. ** Variables considered continuous. ECOG: Eastern Cooperative Oncology Group; PSA: prostate-specific antigen; Hb: haemoglobin concentration; LDH: lactate dehydrogenase; AP: alkaline phosphatase; HR: hazard ratio; CI: confidence interval.

## Data Availability

There is no more data to report.

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
