# Peer review of "Glutamine and Cholesterol Plasma Levels and Clinical Outcomes of Patients with Metastatic Castration-Resistant Prostate Cancer Treated with Taxanes"

_cancers, 2021, doi:10.3390/cancers13194960_

Round 1

Reviewer 1 Report

General: The authors should be congratulated for successfully highlighting that the balance of Glutamine and Glutamate is important for the outcome of castrate-resistant prostate cancer. Overall, the manuscript is easy to read and includes a more detailed exploration of how metabolism/metabolites are affected by the treatment of CRPC. However, there are minor flaws that could be addressed to improve the manuscript further.

Comments:

  • General: please add the time unit (months) in the x-axis of all KM plots.
  • 3 Statistical analysis: please remove the last sentence in the section (left-over from template).
  • Please include the number of events in the baseline characteristics in the study cohort and provide median follow-up time for individual groups to provide an overview of the study cohort early.
  • Page 5: Please check the symbol for micro Molar.
  • General: Please add univariate and multivariate within the COX tables (e.g., Table 2)
  • It would be preferable that Abi/Enza was included in COX analyses (Table 2) since Glutamine is borderline significant for PFS and would be affected by pretreatment of Abi/Enza.
  • The Validation cohort is, per definition, underpowered to perform MVA with 4 levels. A minimum of 10 (often 20 events) is recommended per level. This might explain the missing significance of cholesterol.
  • Figure 5 Panel A: Why are there fewer patients in the OS analysis than PFS/RX-PFS? (84 vs. 81)
  • Figure 5 Panel B) Why are there more patients in the OS analysis than PFS/RX-PFS? (41 vs. 44)
  • Figure S1 A: Is the number at risk for mortality swopped between low and high Glutamate?
  • Figure S4: Why are there two more patients included in the Kaplan-Meier analysis of OS for cholesterol?
  • Why is BMI not included in the analysis, only Statins? Could Glutmaine be a surrogate marker for high BMI and/or metabolic syndrome? Please consider including this in the discussion.

Reviewer 2 Report

The authors present a very interesting manuscript that gives a comprehensive analysis of glutamine metabolism as a novel biomarker and as an interesting regulatory pathway in mCRPC. There are only minor comments that should be addressed by the authors:

Are the patients in the validation cohort mCRPC patients?

Why are the post-chemo samples only available for a proportion of approx. 50% of patients (40/85)? In addition, the missing changes in glutamine levels after taxanes exposure should also be discussed more extensively.
